# Unraveling the Puzzle: Oocyte Maturation Abnormalities (OMAS)

**DOI:** 10.3390/diagnostics12102501

**Published:** 2022-10-15

**Authors:** Safak Hatirnaz, Ebru Hatirnaz, Samettin Çelik, Canan Soyer Çalışkan, Andrea Tinelli, Antonio Malvasi, Radmila Sparic, Domenico Baldini, Michael Stark, Michael H. Dahan

**Affiliations:** 1Mediliv Medical Center, Samsun 55100, Turkey; 2Department of Obstetrics and Gynecology, Samsun Training and Research Hospital, Samsun 55090, Turkey; 3Department of Obstetrics and Gynecology and CERICSAL (CEntro di RIcercaClinicoSALentino), Verisdelli Ponti Hospital, 73020 Scorrano, Italy; 4Department of Biomedical Sciences and Human Oncology, Obstetrics and Gynecology Section, University of Bari “Aldo Moro”, Piazza Aldo Moro, 70100 Bari, Italy; 5Medical Faculty, Clinic for Gynecology and Obstetrics, University Clinical Center of Serbia, University of Belgrade, 11000 Belgrade, Serbia; 6MomòFertilife Clinic, 76011 Bisceglie, Italy; 7The New European Surgical Academy (NESA), 10117 Berlin, Germany; 8McGill University Health Center Reproductive Centre, Department of ObGyn, McGill University, Montreal, QC H3A 0G4, Canada

**Keywords:** oocyte maturation, oocyte maturation abnormalities, empty follicle syndrome, FSH, LH, fertility, stimulation, meiosis, metaphase, meiotic arrest, premature ovarian failure, resistant ovarian syndrome

## Abstract

Oocyte maturation abnormalities (OMAS) are a poorly understood area of reproductive medicine. Much remains to be understood about how OMAS occur. However, current knowledge has provided some insight into the mechanistic and genetic origins of this syndrome. In this study, current classifications of OMAS syndromes are discussed and areas of inadequacy are highlighted. We explain why empty follicle syndrome, dysmorphic oocytes, some types of premature ovarian insufficiency and resistant ovary syndrome can cause OMAS. We discuss live births in different types of OMAS and when subjects can be offered treatment with autologous oocytes. As such, we present this review of the mechanism and understanding of OMAS to better lead the clinician in understanding this difficult-to-treat diagnosis.

## 1. Introduction

Following puberty, dormant ovarian follicles begin to grow through antral and preovulatory stages. Concordantly, oocytes within the follicle develop, mature and ultimately become competent. Oocyte competency normally occurs at the stage of the ovulatory follicle, which can be induced by the hCG trigger in artificial cycles. This resultant maturation occurs with the extrusion of first polar body at the metaphase II stage of meiosis. MII oocytes are generally accepted as mature and competent for fertilization and embryonic development [1]. The resumption of meiosis is the critical step for oocyte maturation. Most of what we understand about human mature oocytes has been extrapolated from animal studies. A high level of cyclic-AMP (c-AMP) is the signal that maintains the oocyte from initiating meiosis. Luteinizing hormone (LH) is the trigger that inhibits the pathways that keep c-AMP at high levels within the oocyte. As such, the LH trigger starts the resumption of meiosis in what was until then an arrested oocyte [2]. Meiotic resumption begins with the germinal vesicle breakdown (GVBD) and is followed by metaphase I and subsequently metaphase II. Oocyte maturation and competence are coordinated during meiotic resumption and maturation [3]. Until the evaluation of the retrieved oocytes from in-vitro fertilization (IVF) cycles, in most cases, no one can determine if oocyte-specific factors are present in patients with infertility. To diagnose oocyte maturation abnormalities (OMAS), the required laboratory findings need to be repeated in at least in two consecutive IVF cycles [3]. Our ongoing studies have revealed that oocytes of patients with OMAS often demonstrate intracycle and intercycle variabilities, with different levels of arrest occurring in different cycles [3], for example, MI arrest in one cycle and germinal vesicle arrest (GV) in a different cycle. Contrary to the present nomenclature that restricts the diagnosis of OMAS to certain subtypes, it seems that the diagnostic evaluation of OMAS needs to be extended. Two classification systems of oocyte maturation arrest were previously published [4,5,6]. These systems are based on excluding other abnormalities [4,5,6]. However, our ongoing study on OMAS revealed that there are novel subsets of oocyte maturation arrest and other pathologies such as oocyte degeneration, empty follicle syndrome (EFS), some forms of premature ovarian failure (POF) and resistant ovarian syndrome (ROS) that could be included in the OMAS classification [3]. Both intrafollicular and extrafollicular factors may cause oocyte pathologies ranging from oocyte degeneration, empty follicles, oocyte maturation arrest and impaired embryonic development [4]. Therefore, there remains confusion on the terminology and classification of the OMAS which still require clarification. Studies conducted on human oocyte maturation and its abnormalities contribute novel information and will clarify the underlying mechanisms. The contemporary approach to the diagnosis of OMAS is to direct couples for oocyte donation without a detailed investigation of the underlying mechanism. However, studies on the genetic basis of OMAS and the first reported live births have focused attention on the possible management of OMAS using the patient’s own oocytes, in certain cases [7]. In this review, we aimed to evaluate and supplement the understanding of oocyte maturation abnormalities with the diagnosis of novel subsets. To comprehend the OMAS, meiotic arrest and resumption must be understood.

## 2. The Orchestration of Meiotic Arrest

Peaking at around 6–7 million follicles at 24 weeks gestational age, oocytes remain in the primordial follicles in a dormant state, arrested at the diplotene phase of the first meiotic division. Most human ovarian follicles undergo apoptosis. Only the follicle which is selected as dominant and undergoes ovulation is prevented from undergoing apoptosis. As such, only about 1–2/1000 follicles have the potential to undergo fertilization in natural cycles [8]. For fertilization, and resumption of meiosis to occur, oocyte maturation and competence are required. Maturation promoting factor (MPF) is a cytoplasmic factor that initiates germinal vesicle breakdown (GVBD) in response to the LH surge in natural cycles or the human chorionic gonadotropin (hCG) or the gonadotropin-releasing hormone (GnRH) analog trigger in artificial cycles [9]. In mouse oocytes, it was reported that cdc25A plays a major role in meiotic resumption, MI spindle formation, and the MI-MII transition [10]. Lincoln et al. studied the role of Cdc25b phosphatase in meiotic resumption in mice. They used Cdc25b knockout mice and reported that they were sterile and oocytes remained arrested at prophase with very limited MPF activity. Understanding the resumption of meiosis depends on the mechanism of meiotic arrest in the follicular unit. When follicles are in the preovulatory stage, the follicle increases intracytoplasmic c-AMP levels and decreases the phosphodiesterase 3A enzyme (PDE3A). The role of the PDE3A enzyme is to cleave c-AMP during meiotic resumption. High levels of c-GMP which are produced in the somatic cells of the follicular unit, block the PDE3A enzyme and increases c-AMP levels contributing to meiotic arrest [9,11]. There is bidirectional communication between the somatic cells made up of the mural granulosa cells and the cumulus granulosa cells and the germ cell (the oocyte). This communication is controlled primarily by the oocyte itself [12,13]. The follicular stimulating hormone (FSH) induces the growth of the antral follicles, induces LH receptor formation and expression, develops the expression of gap junction proteins (connexin 37 and connexin 43), and preserves high c-AMP levels during meiotic arrest [14,15]. Gs proteins are a GTPase that function as cellular signaling proteins and are defined by the alpha subunits they contain. Gs, with its Gs protein-coupled receptor (GPR3), function in the maintenance of the high basal cAMP concentrations during meiotic arrest [16]. In vitro maturation studies give us the opportunity to understand gene expression profiles during oocyte maturation. Yu B et al. studied gene upregulations and downregulations at three human oocyte maturation stages (GV, MI, MII) within the same individual [17]. They reported that DNA methylation plays important roles in gene expression regulation at the different oocyte maturation stages.

## 3. Oocyte Maturation and the Resumption of Meiosis

LH-induced meiotic resumption is a well-known mechanism but highly complex, so many issues remain unresolved. Possible factors that may influence the impaired meiotic resumption may be due to lack of LH activity, defects in the pathways in the follicular unit and or intrinsic (genetic) oocyte factors [1]. Intrafollicular factors that have roles in meiotic resumption and oocyte maturation are as follows: (a) epidermal growth factor (EGF) and its receptor (EGFR); (b) steroids; (c) gap junctions; (d) G proteins; (e) phosphodiesterase; (f) the MAP kinase pathway (MAPK); (g) the NppC/NPR2 pathway. The initial, and one of the most important, steps of meiotic resumption and oocyte maturation is GVBD which is essential for the transition of GV oocytes to the MI stage. Any factors that block this step will result in GV arrest. This is the step of meiotic resumption and MPF activation that prepares the oocyte for the next step of meiosis (MI). Meiotic spindle formation is observed during the MI phase and homologous chromosomes are aligned at the equatorial plate in this phase. Any factors that block the meiotic spindle formation or impair the spindle configuration may result in MI arrest. Following the completion of MI phase, oocyte immediately converts to the next stage of maturation where the first polar body is extruded and MII, the oocytes are physiologically arrested until fertilization occurs. Fertilization then results in further maturation with the extrusion of the second polar body. 

Maturation promoting factor, also known as mitosis promoting factor (MPF), concentrations are high at the GV stage, decline at the MI phase and rise again at the MII transition phase [18]. MII oocyte arrest before fertilization is a physiological process that is mediated by a cytostatic factor (CSF). CSF, functions by either by increasing MPF or by directly blocking the cell cycle regulatory complex (APC/C) (also known as anaphase promoting complex) activity via activating the Emi protein [18]. The Emi protein was reported to be substrate for the MAPK pathway which promotes MII arrest [19].

## 4. Nomenclature of the Oocyte Maturation Abnormalities

The first description of an oocyte factor causing infertility was published in the early 1990s by Rudak et al. This description included four cases that presented with empty follicle syndrome (EFS), GV arrest and MI arrest [20]. EFS is genuine EFS if it is not related to hCG injection or low bioavailability of hCG in the plasma. Levran et al. reported on eight women with unexplained infertility, one with GV arrest, four with MI arrest and three individuals with MII arrest [21]. No information was provided on the subsequent IVF cycle, and whether the MII arrest repeated itself. As such this may be a case of OMAS with not fully matured MII oocytes or just fertilization failure. Hourowitz reported on pregnancies in two women with genuine EFS (G-EFS) by IVM [5]. Beall S et al. published the first classification of oocyte maturation failure soon after Hourowitz et al. They classified OMA into four types (Type I; GV arrest, Type II; MI arrest, Type III; MII arrest and Type IV; mixed arrest) [4]. Galvao et al. studied OMAS and reported the highest number of resistant ovary syndrome (ROS) cases where they obtained five healthy babies after 24 IVM attempts [22] in nine women. ROS was presented as OMAS for the first time. The Hatirnaz and Dahan classification systemin 2019 and the Beall et al. in 2011 classification also both excluded patients with the diagnosis of EFS, premature ovarian failure (POI) and resistant ovary syndrome (ROS). Hatirnaz et al. reported on a large number of OMA cases treated by FSH and hCG primed IVM cycles; however, no pregnancies were reported [6]. In the Hatirnaz and Dahan classification system, GV arrest was defined as Type I, MI arrest was defined as Type II, MII arrest was defined as Type III, GV-MI arrest was defined as Type IV and mixed arrest (GV, MI and MII) was defined as Type V [6].

Further study by Hatirnaz et al. revealed that pregnancies and ongoing pregnancies in women with Type II and Type V OMA can be achieved [7]. Since then, many patients with OMAS were treated at the Hartinaz clinic. This ongoing study gave us the opportunity to try dual stimulation IVM (Duostim IVM) and we were able to see the intracycle variation in oocyte potential [3]. Another finding which was noted was that some cases have a differing subtype of OMAS in subsequent assisted reproductive technology cycles (intercycle variability). Much information related to OMAS is currently lacking and there is much to clarify in future studies. 

Human genetic studies on oocyte maturation abnormalities have contributed significant information on the diagnosis and underlying mechanisms of OMAS. By gathering the findings of our data and those published by others, we have realized that those classification systems previously published are not enough to cover all OMAS diagnoses. Our group has been focused on the diagnosis and treatment of OMAS. This ongoing investigation, which will hopefully soon be published, will allow us to develop novel stimulation protocols in IVM cycles for the management of OMAS. One important observation was that in unstimulated IVM cycles performed in 11 subjects, OMAS was difficult to diagnose due to the not unexpected occurrence of a lack of oocyte maturation after IVM culturing after collection of immature oocytes. Therefore, during IVM cycles the diagnosis of OMAS can only be made in hCG stimulated cycles (unpublished data).

## 5. Types of OMAS

Recent studies using whole genome exomic testing and our ongoing study on treatment options for OMAS confirmed that the current classification systems of OMAS do not reflect the entire spectrum of the disorder. Based on the literature and our recent ongoing research findings, we present all the currently known subtypes in this section. Potential other forms of OMAS were defined as unclassified. The OMAS ranges from (a) oocyte degeneration, (b) empty follicle syndrome, (c) premature ovarian failure, (d) resistant ovary syndrome, (e) oocyte maturation arrest and (f) unclassified forms. 

**a** 
**Dysmorphic and/or Degenerated Oocytes**


Oocyte degeneration and abnormal oocytes can be encountered in the collected oocytes from ART cycles but their occurrence is usually rare. However, some women infrequently experience degeneration of all oocytes retrieved. Oocyte dysmorphism is not uncommon and may be due to extra cytoplasmic or intracytoplasmic factors [23]. The literature suggests that the PANX1 mutation can result in oocyte death [24,25]. Pannexin is a protein molecule playing a role in cellular communication and the PANX1 mutation causes cytoplasmic shrinkage or darkening of oocytes. However, we have one case with PANX1 mutation who had two IVF attempts with GV arrest. With IVM, one bad quality embryo was obtained. We treated her with letrozole priming IVM and obtained 15 oocytes, 4 MII oocytes were injected by ICSI and 1 embryo fertilization was observed without progression. We did not observe oocyte degeneration or death in that particular case. 

A rare form of dysmorphic or degenerated oocyte is necroptosis [26,27]. Necroptosis is a programmed form of inflammatory cell death. It differs from necrosis which is associated with unprogrammed cell death. We have one patient with two previous IVF attempts which resulted in necroptosis. An IVM oocyte pick-up resulted in severely degenerated oocytes at the time of maturation. In a subsequent luteal phase, IVM was completed as part of a duostim IVM cycle, and the collected immature oocytes were evaluated 26 h later and all oocytes were again necroptotic.

**b** 
**Empty Follicle Syndrome (EFS)**


There is a continuous debate on whether EFS is a syndrome or even whether the follicles are really empty. Until recently, it was merely confined to genuine EFS and artificial or false EFS but recent findings and genetic studies have revealed that this is neither a syndrome nor is it follicles empty of oocytes. EFS was first described by Coulan et al. in 1986 in four women with five failed IVF attempts without retrieved oocytes [28]. In the study of Zreik et al., they reported that the incidence of EFS was 1.8% and the recurrence rate was 24% in 34–39-year-old patients and 57% in women over 40 years of age [29]. Early follicular atresia or apoptosis was first mentioned by Uygur et al. as the cause of EFS [30]. Vutyavanich et al. obtained immature oocytes from the filtrate of IVF follicular fluid in a case of EFS [31]. The estimated incidence of EFS ranges from 0.045–7% [32,33,34]. Yakovi et al. defined EFS as genuine EFS and false EFS where a lack of hCG or insufficient timing from the injection until the oocyte collection causes the lack of retrieved oocytes. G-EFS is more prevalent in women with a low number of mature oocytes from IVF cycles [35]. EFS can be a part of OMAS and even can be seen with intercycle variability (unpublished data). This intuitively makes sense since oocytes with few or no LH receptors or even post-receptor defects would be expected to result in immature oocytes with many follicles failing to release the oocytes. Additionally, some women who experienced oocyte degeneration in one cycle can have empty follicles in another cycle. These findings make EFS more complicated and question the diagnosis. EFS may be a part of POI in STAG3 mutation cases [36]. This again makes sense since the oocytes with the most LH receptors would be expected to respond at younger ages while those with less oocyte LH receptors would respond by ovulating later in life and may sub-adequately ovulate. Therefore, EFS should be accepted as part of oocyte maturation abnormalities rather than its own syndrome. The profound luteolytic effect of GnRH analogue trigger in antagonist cycles may result in EFS. In their retrospective study, Castillo et al. reported on the incidence of EFS, which was found to be 3.5% and 3.1%, respectively, of 2034 donation cycles and 1433 own oocyte IVF cycles [37]. EFS can be overcome by the use of IVM, and healthy live births have been reported [5].

**c** 
**Premature Ovarian Failure (POF)/Premature Ovarian Insufficiency (POI)**


POI affecting 1–3% of women aged under 40, manifests itself with the subtypes of OMAS in stimulated IVF cycles. POI is characterized by oligo or amenorrhea and high serum gonadotropin levels [38]. POI varies both phenotypically and genetically [39]. The diverse oocyte profile in women with POI/POF may be due to the novel genetic mutations discovered [40,41,42]. POI is characterized with irregular menstrual dynamics which impairs follicular waves [43]. Apart from other reasons for POI/POF, genetic mutations manifested by ovarian failure are in the range of OMAS. The relationship between POI/POF and genetic mutations remains to be clarified with future studies including WES of cases with POI/POF. Premature ovarian insufficiency (POI), premature ovarian failure (POF) and early menopause are three terms mostly used for the purpose of ovarian ageing; however, all three clinical conditions are separate, though they have similar clinical and laboratory manifestations. POI/POF is defined by the impaired or ceased ovarian function before the age of 40 and is clinically characterized as oligomenorrhea or amenorrhea with high levels of FSH > 25 IU/L and low level of estradiol [44]. POI may be reversible but POF is rapidly progressive condition which results in early menopause [45,46]. POI/POF is very heterogenous but recent studies have revealed that mutations play an important role in the etiopathogenesis [47]. Those who have genetic mutations are the most severe forms of POI/POF [48].

**d** 
**Resistant ovary syndrome (ROS)**


Resistant ovarian syndrome (ROS), also previously known as Savage Syndrome, was first described by Dr. Georgiana Jones in 1969 [49]. Due to FSHR mutations, antral follicular response to both exogenous and endogenous FSH are lacking. The patients are phenotypically and genotypically normal. Typically, FSH and LH levels are elevated but AFC and AMH values are in the normal range if no other additional pathology is associated. Though etiology of ROS is unclear, the most potential reasons seem to be genetic mutations and immunologic [50,51]. ROS is a part of hypergonadotrpoichypogonadizm but differs from POI/POF by the antral follicles present and AMH levels [52].

ROS is not an infrequent form of OMAS where FSH receptor mutations determine the fate of the disease. The receptor ligand connection is impaired and both exogenous or endogenous FSH actions are blocked. FSH and LH levels and AMH levels are high. Both AMH value and AFC of these patients are important in the differential diagnosis from POF [22]. The only way to obtain oocytes in ROS is to perform IVM and retrieved oocytes can be matured in vitro. ROS is an integral part of OMAS and there seems no other way to treat patients with ROS than with IVM. Immunological factors may also play a role in the pathogenesis of ROS [22]. Live births of babies from IVM of ROS have been reported [53,54].

**e** 
**Oocyte Maturation Arrest (OMA, in accordance with Hatirnaz and Dahan classification)**



**GV Arrest (Type I OMA)**


Meiotic resumption is a multistep, complex process including protein production, localization, phosphorylation and degradation [55,56]. Mutations of PATL2 in consanguineous families were found to be related to GV arrest [57]. North American immigrants living in France who were having GV-MI arrest were tested for PATL2 mutation and 26% of the cases were found to have this mutation [58]. A study from China revealed that 44% of GV arrest cases are due to PATL2 mutations [59]. Pure GV arrest is quite a rare entity and we have so far three cases; in one case, we obtained five healthy embryos following a letrozole priming IVM protocol [60,61].


**MI Arrest (Type II OMA)**


The definitive diagnosis of the MI transition is the observation of the extruded polar body 1 (PB1). Here is the most crucial part of meiotic resumption where chromosomal condensation, meiotic spindle formation and alignment take place on the equatorial plate. Both Mei1 and Mlh1 have a role in recombination during the completion of meiosis and knockout mice studies showed that their absence result in MI arrest [62]. The TUBB8 mutation was first reported to cause MI arrest in 2016. TUBB8 is human specific tubulin protein which is essential for proper formation of meiotic spindles [63]. Either dominant or recessive inherited patterns of TUBB8 mutations are pathogenic and almost 30% of all MI arrest cases carry different types of TUBB8 mutations [64,65,66,67]. Bisphenol A (BPA), commonly used in the IVF lab as plastic materials, can damage spindle configuration and chromosomal alignment and can cause MI arrest at high doses in mouse oocytes [68]. Biallelic missense mutation of TRIP13 was found to cause MI arrest. Injection of TRIP13 cRNA to affected individuals rescued the oocyte from MI arrest, which may be an opportunity to overcome MI arrest due to TRIP13 mutation [69].


**MII Arrest (Type III OMA)**


MII oocyte is the last arrest point of meiotic resumption before fertilization and physiological arrest is a must for further progress. Physiological MII arrest is regulated by cytostatin factor (CSF). MPF concentration reaches a maximum level at the MII stage and then mature oocyte is arrested at this phase. CSF stabilizes MPF and keeps chromosomes condensed, thus a second round of DNA replication during transition from MI to MII is avoided [70,71]. It is the fertilization of the MII oocyte that completes the meiotic resumption by extruding the second polar body (PB2). Not all MII oocytes are functionally competent though they are configurationally MII and mature. Morphologically normal MII oocytes may show fertilization failure. MII arrest is the rarest form of OMAS and we have seen only two cases. However, they have not yet been treated. The difference between fertilization failure (FF) and MII arrest is that FF is not a repeated process but MII arrest is repeated in each cycle. MII arrest requires increased synthesis and accumulation of cyclin B. Meng et al. studied the role of cyclin B (Ccnb3) on MII arrest in mouse oocytes and showed that the gradual decrease in Ccnb3 is required for meiotic maturation [72]. Ccnb3 participates in the separation of homologous chromosomes during the first meiotic process by forming a complex with cyclin dependent kinase (CDK1). Fertilization is a great transformation period where oocyte is transformed to the embryo and meiotic division is transformed to mitotic division. The diagnostic determinants of fertilization are sperm egg binding, pronuclear bodies (PNs), PB2 and the presence of cortical granules. Knockout of genes encoding Juno, CD), Izuma, Adam2, Adam3 and Calmegin in sperm produce fertilization failure [73,74,75]. A homozygous single gene mutation of TLE6 in three infertile patients in two consanguineous families is reported to be the reason for fertilization failure in morphologically normal appearing oocytes [76]. Homozygous WEE2 mutation in four families were reported to be causative of FF. This system indirectly impairs Cdc2 function [77]. Homozygous mutation and compound heterozygous mutation in five cases of CDC20 were reported to be the reasons for FF [78].


**GV and MI Arrest (Type IV OMA)**


This subtype is one of the mostly encountered form in our OMAS study. We added this as a subtype to Hatirnaz and Dahan Classification and we believe that this subtype needs to be investigated in more detail. GV-MI maturation arrest is related to MutL homolog 3 (Mlh3) and ubiquitin b (Ubb) proteins in mice. Mlh3 functions in both DNA mismatch repair and meiosis. *Mlh3*–/–oocytes fail to complete meiosis I after fertilization. Mlh3 has an essential and distinct role in mammalian meiosis. Deficiency of Ubb, a member of ubiquitin family causes GV and MI arrest in mice [79,80]. *Ubb*^_/_^ oocytes do not proceed beyond metaphase I. Loss of one Ubb family member may be compensated by the expression of other Ubb family members (Uba 52, Uba80 though this compensation is not enough to successfully compete the meiotic resumption). Human oocytes try to overcome the impact of oocyte maturation arrest by compensatory mechanisms that have yet to be clarified. This may be succeeded by the over-expression of other family members of the mutated proteins, wildtype expression or by making the mutated gene under expression.


**Mixed Arrest (Type V OMA)**


Arrest at the GV-MI-MII oocyte stages in one individual can be seen in routine IVF cases and accepted as normal if the number of MII oocytes that end up fertilizing are large enough that complete FF does not occur. However, there are some cases where in their repeated cycles, GV-MI-MII oocytes were retrieved but the number of MII is less than 25% of all oocytes retrieved. Additionally, fertilization failure or early and late embryonic arrest occurs. This subtype is the most common form that we have faced. Literature from animal studies showed that deficiency of Mhl3, which belong to a family including Mlh1, Pms1 and Pms2, results in MI and MII arrest [79]. Whether it is the decelerated meiotic resumption or the compensation of meiotic arrest remains to be clarified in future studies. MII oocytes in this subtype may be morphologically mature but look incompetent when the fertilization and embryonic course is evaluated. Zygotic cleavage failure (ZCF) is a novel nomenclature and is seen in this subtype. In ZCF, factors related to sperm mutations may also play a role in the pathogenesis, which would be unrelated to the oocyte maturation. Homozygous mutations in BTG4 (B cell translocation gene 4) are reported to cause ZCF [81].


**f) Unclassified types which also exist (with causes which remain unknown)**

**f-1) Empty zona-GV arrest**

**f-2) GV-MII arrest**

**f-3) MI and MII arrest**


In our ongoing study on OMAS, we found cases whose cycles revealed empty zona-GV arrest in one cycle and GV-MII arrest in another. In addition, we have three cases with MI-MII arrest, and obtaining embryonic development in this unclassified OMAS is very difficult. Early embryonic cleavage arrest was observed in the sole case with fertilization. 


**Issues left to ponder**


It remains to be understood why different forms of oocyte maturation abnormalities are seen in some cases and what the mechanism of intracycle variability are. One case admitted to our center for IVM for OMAS was diagnosed with an FSH receptor mutation. The patient’s history is unique and deserves discussion. She had GV-MII oocytes collected in her first IVF cycle with fertilization failure. Empty zona-GV arrest was noted in her second attempt. We recommended mutation analysis specific for oocytes and determined an FSH receptor mutation. We performed three IVM cycles but failed to achieve embryonic development. The patient was stimulated for IVF in another center and embryonic development was achieved which resulted in an ectopic pregnancy. This suggests that some mechanism may exist to compensate for FSH receptor mutations. Other factors, such as autoimmunity, increased expression of some local factors, including TNF-alpha, and other intrafollicular factors may also play roles in the development of OMAS.

## 6. Conclusions

Dealing with OMAS is difficult. OMAS presents areas of research for geneticists, clinicians and bench researchers. Directing women with OMAS to oocyte donation is the easiest way to obtain a pregnancy. However, this may be undesirable for some couples. More patient data are needed to understand the reasons which cause OMAS. Oocytes must be evaluated by electron microscope and fluorescent microscope. The follicular fluid content of OMAS cases should be investigated for TNF-alpha, gene expressions, local factors, wildtype gene expressions, and other potential factors that may cause meiotic arrest as well as those factors that compensate for the OMAS inducing defects. Much remains to be learned about OMAS. It should be remembered that certain women with OMAS can achieve pregnancies with autologous gametes using IVM.

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
