# Peer review of "Unraveling the Puzzle: Oocyte Maturation Abnormalities (OMAS)"

_diagnostics, 2022, doi:10.3390/diagnostics12102501_

Round 1

Reviewer 1 Report

Summary

The authors have written a review article on oocyte maturation arrest (OMAS) to highlight some limitations of the current classification of OMAS and the importance of being able to more accurately diagnose the causes of OMAS to be able to offer treatment options.

A search for review articles on “oocyte maturation arrest” resulted in only 4 articles; two recent broader articles on genetic factors related to infertility including oocyte maturation arrest and one more specific review on possible mechanisms of maturation arrest from 2003. Therefore, a review of this topic is timely.

However, the manuscript could be significantly improved by providing clear definitions of the terms and more detailed information. Some changes are recommended.

The manuscript requires further editing to improve the grammar. Some examples from early in the manuscript are provided in the minor comments section below.

Specific comments

Line 58 to 59 states “Two classification systems of oocyte maturation arrest were published by Hourwitz” but the authors have included 3 references to papers by other authors. It is not clear what these classification systems are.

The definition of OMAS needs to be clearly stated. Is there a difference between OMAS and OMA? If not then it would be clearer for the reader if one term or the other is used consistently. Lines 150 and 157 use OMA, elsewhere uses OMAS.

Line 54 the authors state patients with OMAS often demonstrate intracycle variabilities, with different levels of arrest occurring in different cycles. Should this be “intercycle” variabilities instead of “intra”? Or do you mean both intra and intercycle variation? Please clarify this.

Lines 109-111 makes the point that 1,077 genes were upregulated and 3,758 were downregulated in transition from MI to MII. Can you please expand on the significance of this statement and perhaps give some examples of the genes?

Lines 132-138 discuss CSF and Emi protein with reference to 2005 and 2004 review papers. Can you provide more recent references from original studies? For example, have these factors been identified?

Figure 1 needs a full caption describing what the figure is showing, including what the abbreviations refer to. Many of the factors shown in this figure have not been mentioned in the text. These should either be discussed in the text or the figure changed accordingly.

Section 4 Nomenclature

·      Please explain what is meant by genuine EFS as opposed to EFS.

·      Define resistant ovary syndrome (ROS).

·      Line 154 the reference for the Hatirnaz and Dahan classification is missing.

·      A key point made here is the authors observation of intracycle and intercycle variation in oocyte potential, but it would be helpful to provide a summary of what varied in the text, without requiring the reader to read the references.

·      Lines 176-180 The authors state that in the unstimulated “IVM” cycles OMAS was difficult to diagnose due to a lack of mature oocytes at collection. Do you mean that there were a limited number of oocytes collected, or none were mature at collection, or that they did not mature during IVM? I would have thought that if it is a natural cycle and ovulated oocytes were collected that were immature these could be classified as arrested oocytes. Or if they underwent IVM and didn’t mature, they could also be classified as arrested oocytes. Please clarify this.

Section 4 Types of OMAS – this is misnumbered

·      Line 184 refers to “our recent research findings” – do you mean published findings (in which case you should provide the reference here) or the unpublished data mentioned in the previous section?

·      I don’t think Figure 2 is very helpful. If it is included, it needs further explanation in the legend and some of the text needs to be enlarged to make it legible. It is not clear what the purpose of the bottom part of the figure is; are these OMA denuded IVF oocytes different to OMA IVM oocytes above? The legend references number 3. Perhaps clarify that this figure is an interpretation of the data from that reference rather than a figure taken from it. Also, the definitions of the abbreviations need to be provided in the figure legend.

c) Premature Ovarian Failure

This section would benefit from a little more explanation. For example, the authors state that POI varies both phenotypically and genetically. Can you provide some examples of how it varies and then link this to which ones correspond to OMAS?

d) Resistant ovary syndrome

Similarly, this would benefit from a little more description. It hasn’t been well-defined earlier in the manuscript.

References

There are some errors in the format and numbering of these that need to be corrected.

Minor comments - examples

Abstract line 21 – put OMAS in brackets

Line 27 – “prevent” should be “present”

Line 29 – perhaps “diagnosis” should be “syndrome”

Line 53 “in at least in two consecutive..”

Line 58 “need” should be “needs”

Lines 102-103 – mixed singular/plural – perhaps change to “Gs proteins are GTPases that function..... and are defined by the....”

Line 107-108 “gene up- and down-regulation”

Lines 127 to 130 – “Following the completion of the MI phase, the oocyte immediately.... extruded, and the resulting MII oocyte is normally arrested...”

Line 284 – avoid using non-English language

Line 304 – is this meant to say “contagious”?

Line 316 – correct Ubb-/-

Author Response

Line 58 to 59 states “Two classification systems of oocyte maturation arrest were published by Hourwitz” but the authors have included 3 references to papers by other authors. It is not clear what these classification systems are.

Reply to reviewer; The sentence was rewritten, corrected and highlighted in the manuscript

The definition of OMAS needs to be clearly stated. Is there a difference between OMAS and OMA? If not then it would be clearer for the reader if one term or the other is used consistently. Lines 150 and 157 use OMA, elsewhere uses OMAS.

Reply to reviewer; OMA stands for oocyte maturation arrest which is now in the range of OMAS which is oocyte maturation abnormalities. OMA is correct in lines 150-157.

Line 54 the authors state patients with OMAS often demonstrate intracycle variabilities, with different levels of arrest occurring in different cycles. Should this be “intercycle” variabilities instead of “intra”? Or do you mean both intra and intercycle variation? Please clarify this.

Reply to reviewer; Reviewer is absolutely right both intracycle and intercycle variabilities observed and this is changed and highlighted in the text.

Lines 109-111 makes the point that 1,077 genes were upregulated and 3,758 were downregulated in transition from MI to MII. Can you please expand on the significance of this statement and perhaps give some examples of the genes?

Reply to reviewer; For diminishing the wordcount of the manuscript we minimalize the content of the text and deletion of this sentence looks better instead of adding gene expressions. The sentence in line 109-111 is deleted.

Lines 132-138 discuss CSF and Emi protein with reference to 2005 and 2004 review papers. Can you provide more recent references from original studies? For example, have these factors been identified?

Reply to reviewer: Pubmed search revealed the most recent version of CSF/Emi protein was 2006 (Schmidt A et al,J Cell Sci,2006) and not adding more to the ones we cited as references.

Figure 1 needs a full caption describing what the figure is showing, including what the abbreviations refer to. Many of the factors shown in this figure have not been mentioned in the text. These should either be discussed in the text or the figure changed accordingly.

Reply to reviewer; The figure includes many mechanisms in it and abbreviations are for pathways involved or intrafollicular factors. We add this figure to show the mechanism of meiotic arrest and resumption. Removing this figure will not change the text integrity thus we deleted the figure 1 from the main text

Section 4 Nomenclature

Please explain what is meant by genuine EFS as opposed to EFS.

Reply to Reviewer; An explanation about g-EFS is added and highlighted

  • Define resistant ovary syndrome (ROS).

Reply to Reviewer; ROS is defined in detail in line 247-254

  • Line 154 the reference for the Hatirnaz and Dahan classification is missing.

Reply to Reviewer;(6) is marked in the text and highlighted

  • A key point made here is the authors observation of intracycle and intercycle variation in oocyte potential, but it would be helpful to provide a summary of what varied in the text, without requiring the reader to read the references.

Reply to Reviewer; Intracycle and intercycle variability is the observation of different levels of oocyte maturation abnormalities in patients treated with Duostim IVM and example of the variation was writ ten in lines 53-57. This is a novel observation deserves mentioning in the main text,

  • Lines 176-180 The authors state that in the unstimulated “IVM” cycles OMAS was difficult to diagnose due to a lack of mature oocytes at collection. Do you mean that there were a limited number of oocytes collected, or none were mature at collection, or that they did not mature during IVM? I would have thought that if it is a natural cycle and ovulated oocytes were collected that were immature these could be classified as arrested oocytes. Or if they underwent IVM and didn’t mature, they could also be classified as arrested oocytes. Please clarify this.

Reply to Reviewer; The sentence is rewritten and highlighted actually the sentence is for explaining the uselessness of unstimulated IVM in the diagnosis of OMAS

Section 4 Types of OMAS – this is misnumbered

Reply to Reviewer; There is no misnumbering of the types of OMAS only f was absent and it is added and highlighted in the text

  • Line 184 refers to “our recent research findings” – do you mean published findings (in which case you should provide the reference here) or the unpublished data mentioned in the previous section?

Reply to Reviewer; ongoing research is added and highlighted

  • I don’t think Figure 2 is very helpful. If it is included, it needs further explanation in the legend and some of the text needs to be enlarged to make it legible. It is not clear what the purpose of the bottom part of the figure is; are these OMA denuded IVF oocytes different to OMA IVM oocytes above? The legend references number 3. Perhaps clarify that this figure is an interpretation of the data from that reference rather than a figure taken from it. Also, the definitions of the abbreviations need to be provided in the figure legend.

Reply to Reviewer; This figure is the heart of the manuscript that summarizes and partly classifies the whole study and our findings in our ongoing study. Abbreviations were added at the end of the text. OMA denuded IVM oocytes means that those oocytes of OMA were IVM oocytes where we diagnose the patients and validate them when compared to IVF oocytes.OMA IVF denuded oocytes mean that those oocytes were IVF oocytes of the patients in their previous IVF cycles not IVM oocytes of our study. The legend referred to literature number 3

  1. c) Premature Ovarian Failure

This section would benefit from a little more explanation. For example, the authors state that POI varies both phenotypically and genetically. Can you provide some examples of how it varies and then link this to which ones correspond to OMAS?

Reply to Reviewer; authors enriched the POI/POF session by adding two paragraphs and 4 new references and highlighted in the text

  1. d) Resistant ovary syndrome

Similarly, this would benefit from a little more description. It hasn’t been well-defined earlier in the manuscript.

Reply to Reviewer: We enriched the ROS session accordingly and highlighted in the text

References

There are some errors in the format and numbering of these that need to be corrected.

Reply to Reviewer; they were corrected.

Minor comments - examples

Abstract line 21 – put OMAS in brackets

Reply to reviewer; It was corrected.

Line 27 – “prevent” should be “present”

Reply to reviewer; It was changed.

Line 29 – perhaps “diagnosis” should be “syndrome”

Reply to reviewer; That sentence is correct in that form.

Line 53 “in at least in two consecutives...”

Reply to reviewer; It was corrected.

Line 58 “need” should be “needs”

Reply to reviewer; It was changed.

Lines 102-103 – mixed singular/plural – perhaps change to “Gs proteins are GTPases that function..... and are defined by the....”

Reply to reviewer; It was corrected.

Line 107-108 “gene up- and down-regulation”

Reply to reviewer; It was corrected.

Lines 127 to 130 – “Following the completion of the MI phase, the oocyte immediately.... extruded, and the resulting MII oocyte is normally arrested...”

Reply to reviewer; It was corrected.

Line 284 – avoid using non-English language

Reply to reviewer; It was changed and highlighted

Line 304 – is this meant to say “contagious”?

Reply to reviewer; It is changed as consanguineous and highlighted in the text

Line 316 – correct Ubb-/-

Reply to reviewer; It was corrected.

Reviewer 2 Report

This review article addresses the different aspects of folicular failure that lead to infertility. Aspects referring to types of abnormality as well as their classification are adequately addressed. The proposal for a new classification is also interesting as it will contribute to the diagnosis and eventual treatment of the alterations. However, for the effect of AOM to be understood, a coherent review of the intrafollicular mechanisms of blocking and resumption of meiosis is important. The article fails in this part. The description of intrafollicular physiological mechanisms is confusing and does not help the reader to understand the rest of the text. All comments were made directly in the article

Author Response

In this review the authors aimed to update the classification of oocyte maturation abnormalities. Despite the goal is very interesting and the manuscript quite clear the reason why some unclassified cases of OMAS are included in certain groups are not clear and/or properly explained. The manuscript is nice to read at the beginning but then it became redundant and poor of convincing explanations. At the end some clinical cases are just cited. I agree with the fact that many unknow aspects need to be investigated but the manuscript should better guide the reader to understand why this new classification is proposed and which are the scientific hypothesis of certain choices.

Reply to reviewer; Line 92 is corrected and highlighted. Figure 1 was deleted. For the understanding of the arrest mechanism a very long paragraph together with many citations are needed. Our primary objective is to present the OMAS not to discuss maturation arrest and resumption in detail. We also deleted the genetic components from this manuscript to adapt the paper for Diagnostics journal.

Reviewer 3 Report

In this review the authors  aimed to update the classification of oocyte maturation abnormalities. Despite the goal is very intersting and the manuscript quite clear the reason why some unclassified cases of OMAS are included in certain groups are not clear and/or properly explained. The manuscript is nice to read at the beginning but then it become redondant and poor of convincing explanantions. At the end some  clinical cases are just cited. I agree with the fact that many uknow aspects need to be investigated but the manuscript should better guide the reader to understand why this new classification is proposed and which are the scientific hypothesis  of certain choises.

Author Response

In this review the authors aimed to update the classification of oocyte maturation abnormalities. Despite the goal is very interesting and the manuscript quite clear the reason why some unclassified cases of OMAS are included in certain groups are not clear and/or properly explained. The manuscript is nice to read at the beginning but then it became redundant and poor of convincing explanations. At the end some clinical cases are just cited. I agree with the fact that many unknow aspects need to be investigated but the manuscript should better guide the reader to understand why this new classification is proposed and which are the scientific hypothesis of certain choices.

Reply to reviewer; Primary objective of this manuscript is not to present a new classification system, rather to present that previous classification systems used were not covering the whole scenario. The figure we presented is to show that there are differences in the oocytes retrieved either in the same cycle or in the next cycles. This is the field that is absolutely bypassed by the investigators since there is the opportunity to offer oocyte donation. This is why the information is lacking and most of the issues are hypothetical and the paper looks redundant in some aspects. General approach for cases with OMA is to classify the cases by excluding POF, ROS, EFS and oocyte degenerations. However, with the gathering human mutation data, it seems that OMAS need to replace OMA. Considering the number of patients in one center (so far 55 OMAS cases which is a big number for such cases), we thought to mention about the novel cases as unclassified.

Round 2

Reviewer 1 Report

The authors have improved the manuscript. There are still a few minor corrections to make:

Figure 2 should now be Figure 1 and I still think the font in the figure is too small. "Ethiology" needs to be corrected to "etiology".

There are still two sections numbered "4."

Line 243 What does "WES" stand for? (has it been defined elsewhere in the text)

Line 52-53 still needs correcting to "in at least two consecutive IVF cycles"

Author Response

"Etiology" needs to be corrected to "etiology".

Answer: it was corrected.

Figure 2 should now be Figure 1 and I still think the font in the figure is too small. Answer: Figure 2-1 was removed from the text, since we could not get permission from the previous journal to use .

There are still two sections numbered "4." 

Answer: There is a typo. there are two paragraphs numbered the same way, with the number 4. Paragraphs have been renumbered.

Line 243 What does "WES" stand for? (has it been defined elsewhere in the text).

Answer: WES stands for whole genome exomic study. It was changed.

Line 52-53 still needs correcting to "in at least two consecutive IVF cycles" .

Answer: It was corrected.

Reviewer 3 Report

The revised manuscript is improved.

Author Response

Thnaks for reviewer positive comment.